# Road Extraction from High-Resolution Remote Sensing Imagery Using Refined Deep Residual Convolutional Neural Network

**Lin Gao [1] [ID], Weidong Song [1],*, Jiguang Dai [1] and Yang Chen [2] [ID]**

[1] School of Geomatics, Liaoning Technical University, Fuxin 123000, China; gaolin19920324@163.com (L.G.); 13841800773@163.com (J.D.)

[2] State Key Laboratory of Information Engineering in Surveying, Mapping and Remote Sensing, Wuhan University, Wuhan 430079, China; chenyang1017@126.com

* Correspondence: Songweidong@lntu.edu.cn

**Abstract:** Road extraction is one of the most significant tasks for modern transportation systems. This task is normally difficult due to complex backgrounds such as rural roads that have heterogeneous appearances with large intraclass and low interclass variations and urban roads that are covered by vehicles, pedestrians and the shadows of surrounding trees or buildings. In this paper, we propose a novel method for extracting roads from optical satellite images using a refined deep residual convolutional neural network (RDRCNN) with a postprocessing stage. RDRCNN consists of a residual connected unit (RCU) and a dilated perception unit (DPU). The RDRCNN structure is symmetric to generate the outputs of the same size. A math morphology and a tensor voting algorithm are used to improve RDRCNN performance during postprocessing. Experiments are conducted on two datasets of high-resolution images to demonstrate the performance of the proposed network architectures, and the results of the proposed architectures are compared with those of other network architectures. The results demonstrate the effective performance of the proposed method for extracting roads from a complex scene.

**Keywords:** refined deep residual convolutional neural network; road extraction; remote sensing; tensor voting; math morphology; high-resolution imagery

## 1. Introduction

Roads play a key role in the development of transportation systems, including the addition of automatic road navigation, unmanned vehicles, and urban planning, which are important in both industry and daily living [1]. Automatic road extraction from high-resolution optical remote sensing imagery is a fundamental task [2].

However, road extraction from high-resolution images has two challenges: (i) The images have complex road structures; remote sensing images are usually characterized by complexity in the form of heterogeneous regions with large intraclass variations and lower interclass variations [3]. The heterogeneity in remote sensing images restricts many existing algorithms that depend on a set of predefined features extracted using tunable parameters. (ii) The objects in images are blocked by obstacles, either through shadow occlusion or visual occlusion. Roads can be roughly recognized from images with shadow occlusion, but those with visual occlusion cannot reflect road information. The shadows of trees or buildings on the roadsides and the vehicles on the roads can be observed from high-resolution imagery [4], as shown in Figure 1.

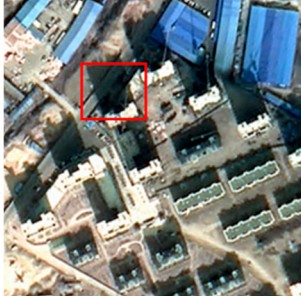 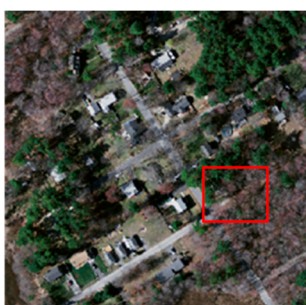

**Figure 1.** Instances of roads blocked by obstacles (red rectangles). **Left**: shadow occlusion; **Right**: visual occlusion.

As indicated by the extensive research in the literature, some researchers have used conventional methods or machine learning algorithms to solve these challenges. A semiautomatic method was proposed using mean shift to detect roads. The method extracted the initial point from road seed points using a threshold to separate the boundary between roads and non-roads [5]. Probability and graph theory have been used to extract road networks [6]. Machine learning algorithms are generally more accurate than the abovementioned methods [7]. For example, Song and Civco [8] proposed a method that utilized shape index features and support vector machines (SVMs) to detect road areas. Das et al. [9] exploited two salient features of roads and a multistage framework designed to extract roads from high-resolution multispectral images. Alshehhi and Marpu [10] proposed an unsupervised road extraction method based on hierarchical graph-based image segmentation.

In recent years, deep learning has been a popular research topic because it can mine high-level features and has improved the effectiveness of many computer vision tasks. Methods based on deep convolutional neural networks have achieved state-of-the-art performance on a variety of computer vision tasks [11], such as classification [11–14], semantic segmentation [15–17], object detection [18,19] and other applications [20,21]. These methods provide better results than conventional methods in terms of the first challenge and shadow occlusion problem. In the field of road extraction, Mnih and Hinton [22] proposed a method that employed restricted Boltzmann machines (RBMs) to detect road areas from high-resolution aerial images. A preprocessing step before detection and a postprocessing step after detection were used. Preprocessing was deployed to reduce the dimensionality of the input data. Postprocessing was employed to remove disconnected blotches and fill in the holes in the roads. Satio et al. [23] employed convolutional neural networks to extract buildings and roads directly from raw remote sensing imagery. Ramesh et al. [24] designed a U-shaped FCN for road extraction by using a stack of convolutions followed by deconvolutions with skip connections. Zhang et al. [1] combined deep residual networks (ResNets) [25] and U Net [26], which allowed networks to be designed with few parameters but improved results. Alexander V. Buslaev et al. [27] proposed a fully convolutional neural network of the U-Net family via ResNet34 and decoding adapted from vanilla U-Net.

In this paper, we propose a novel method for road extraction from high-resolution imagery. The method is a refined deep residual convolutional neural network (RDRCNN) framework with a postprocessing stage. RDRCNN is a symmetric framework consisting of two major units: the residual connected unit (RCU) and the dilated perception unit (DPU). Compared to the existing methods, the proposed method exploits texture information that exhibits high-level features. This information improves extraction decisions without the need for any manual specific spectral information process because a pretrained network can extract rich and distinct high-level representations for visual objects in images.

The proposed method makes the following three main contributions.

(1) A new dataset is provided from GF-2 high-resolution satellite images. The resolution is 0.8 m.

(2) An RDRCNN structure is introduced for road extraction using high-resolution remote sensing imagery.

(3) A novel postprocessing method based on a math morphology and tensor voting algorithm is used to integrate broken roads and improves the performance due to the connectivity of roads.

## 2. Materials and Methods

In this section, the proposed framework for extracting roads in high-resolution imagery is illustrated. The method does not require any preprocessing stage. First, the RDRCNN architecture and basic unit framework are discussed. Second, a tensor voting algorithm is used to alleviate the problem of local broken roads and enhance the outputs at the postprocessing stage.

### 2.1. The Structure of the Refined Deep Residual Convolutional Neural Network

The RDRCNN architecture is an end-to-end symmetric training structure to predict pixel-level results and was inspired by ResNet [25], U Net [26], and Deeplab [28]. RDRCNN consists of two core units, including an RCU and a DPU, followed by a full convolution layer.

The architecture is designed with three parts, as shown in Figure 2. The first part is designed to extract the features using some RCUs (the blue blocks shown on the left of Figure 2) with a shrinking structure via some max-pooling operators. The second part at the bottom of Figure 2 (orange blocks) is for enlarging the field of view (FOV) without losing resolution by using consecutive multi-scaled dilated connected units. The third part is an extensive structure for generating a road extraction map that is the same size as the input.

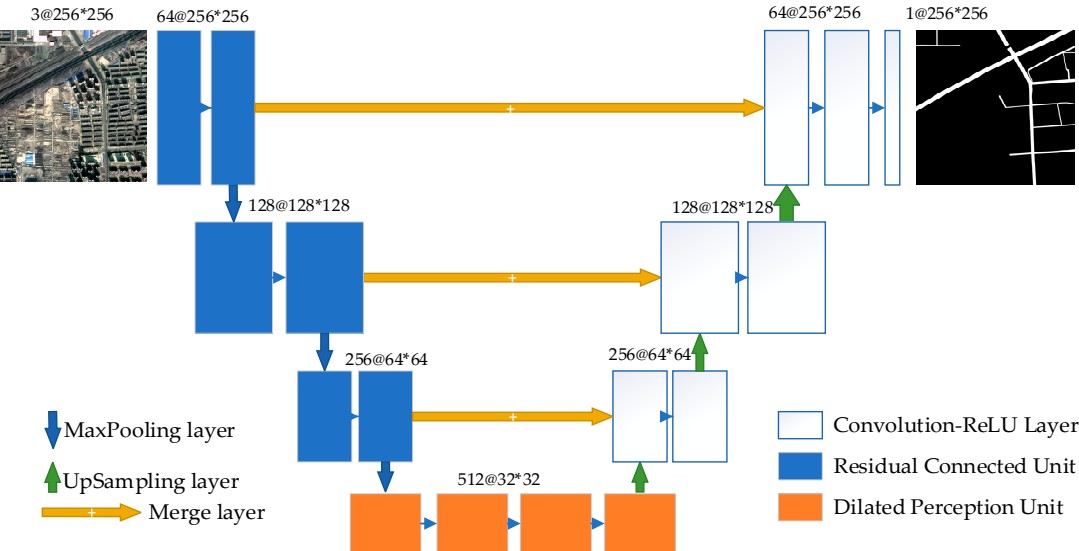

**Figure 2.** Illustration of the refined deep residual convolutional neural network (RDRCNN).

The basic components of each unit usually consist of different operators, including convolution (dilated convolution, full convolution [29], etc.), nonlinear transformation (ReLU, sigmoid, etc.), pooling (max-pooling, average-pooling, etc.), dropout, concatenate and batch normalization. The convolution produces new features, each element of which is obtained by computing a dot product between the FOV and a set of weights (convolutional kernels). The convolution layer needs to be activated by a nonlinear function to improve its expression capability. However, the repeated combination of max-pooling and striding at consecutive layers of these networks significantly reduces the spatial resolution of the resulting feature maps. A partial remedy is to use the up-sampling layer, which requires additional memory and time. Because of the characteristics of remote sensing imagery (e.g., large cover regions, high resolution, complex backgrounds), a deeper network structure can

theoretically gain more effective information for the goal task. However, the exploding gradient problem and vanishing gradient problem may occur [30]. Therefore, an RCU and a multi-scaled DPU based on [28] and [31] are used to alleviate these problems, and these units are discussed as follows.

### 2.1.1. Residual Connected Unit

Residual learning and identity mapping by shortcuts were first proposed in [25]. This procedure exhibits good performance in the field of computer vision. In [25], the building block was defined by the equation below:

$$x_{l+1} = \omega_{l+1}\sigma(\omega_l x_l) + x_l \tag{1}$$

where $x_{l+1}$ and $x_l$ are the output and input vectors of the considered layers, respectively. The function $\sigma(\cdot)$ represents ReLU [32], and the biases are omitted to simplify the notations. Equation (1) is performed by a shortcut connection and elementwise addition. Then, the second activation function is followed by the addition. He et al. [25] presented a detailed discussion of the impacts of different combinations and suggested a full pre-activation design. In this paper, the shortcut connected unit is modified as shown in Figure 3, whose details are depicted in Table 1. To prevent the overfitting gap, we add the batch normalized layer [33] to the bottom of the basic unit.

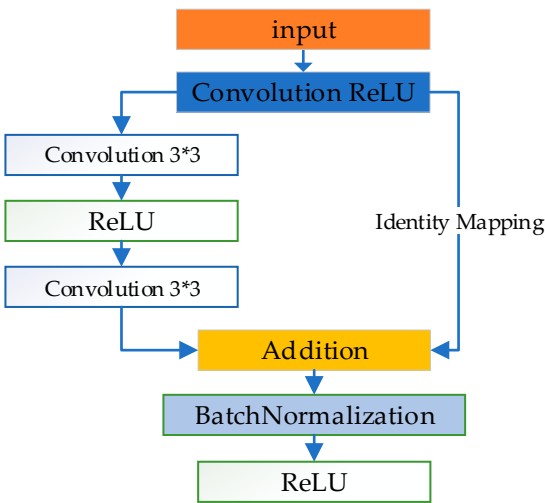

**Figure 3.** The structure of residual connected unit.

**Table 1.** The details of Residual connected unit.

| Items | Layer | Kernel Size |
|---|---|---|
| RC-1 | Conv-ReLU | $1 \times 1$ |
| RC-2 | Conv-ReLU | $3 \times 3$ |
| RC-3 | Conv | $3 \times 3$ |
| RC-4 | Addition | - |
| RC-5 | Batch Normalization | - |
| RC-6 | ReLU | - |

### 2.1.2. Dilated Perception Unit

To satisfy both the large receptive field and the high spatial resolution, we adopt dilated convolution [31]. The dilated convolutions enlarge the receptive field while maintaining the resolution [34]. As shown in Figure 4, a DPU consists of the dilated convolution layer and the full convolution layer. The former utilizes specific kernels with sparsely aligned weights to enlarge the FOV, and the latter retains the relationships among the neighborhood. Both kernel size and the

interval of sparse weights expand exponentially with the increase in the dilation factor. By increasing the dilation factor, the FOV also expands exponentially [35].

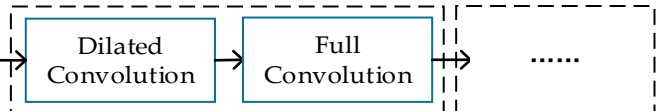

**Figure 4.** Illustration of the dilated perception unit structure.

In this paper, we design four scales for the DPUs, as done in a previous study [31]. Details of the experimental parameters are described in Table 2.

**Table 2.** Details of dilation convolution units with different dilated scales.

| Items | Kernel Size | Numbers of Feature Maps | Dilated Scale |
|-------|-------------|-------------------------|---------------|
| DPU-1 | $3 \times 3$ | 512 | 6 |
| DPU-2 | $3 \times 3$ | 512 | 12 |
| DPU-3 | $3 \times 3$ | 512 | 15 |
| DPU-4 | $3 \times 3$ | 512 | 24 |

### 2.2. Postprocessing

RDRCNN detects road regions but does not guarantee continuous road regions, especially around road intersections. However, it can lead to broken roads that were blocked by shadows or trees in the RDRCNN outputs. To solve address this disadvantage, a postprocessing step is used to reduce broken regions and improve topology expression.

In this work, the tensor voting (TV) algorithm [36], which is a blind voting method between voters, is used in postprocessing. In this paper, the algorithm implements the smoothness constraint to generate descriptions in terms of regions from RDRCNN outputs. The method is based on tensor calculus for representation and linear voting for communication [37].

- Encoding the RDRCNN outputs into tensors

First, a second-order symmetric semipositive tensor is used to encode the data of the predicted structures in the input image. In this section, these tensors can be visualized as ellipses [37]. A second-order symmetric tensor T in 2D can be represented as a nonnegative definite $2 \times 2$ symmetric matrix, which can be generated by the following equation:

$$T = \lambda_1 e_1 e_1^T + \lambda_2 e_2 e_2^T = (\lambda_1 - \lambda_2) e_1 e_1^T + \lambda_2 \left( e_1 e_1^T + e_2 e_2^T \right) \tag{2}$$

where $e_1$ and $e_2$ are the eigenvectors of $T$, and $\lambda_1$ and $\lambda_2$ are their respective eigenvalues. $e_1 e_1^T$ describes a stick tensor, and $\lambda_2 \left( e_1 e_1^T + e_2 e_2^T \right)$ describes a ball tensor.

- Fundamental stick voting field and stick votes

Second, while voting, tensors cast votes in different positions along the space. The votes constitute the voting field of a tensor at every location. During TV, the direction of each tensor is cast as a vote at a certain site that has a normal along the radius of the osculating circle that connects the voter with the vote location, as illustrated in Figure 5. This condition comes out of the observation that the osculating circle represents a smooth continuation of an oriented feature. The received vote is rotated at an angle of 2θ with respect to the voter. The original formulation crops the votes at outside locations $(-\pi/4 \le \theta \le \pi/4)$.

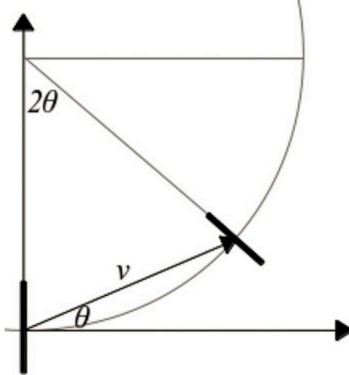

**Figure 5.** Direction of a stick vote pointing to the center of the osculating circle [37].

The vote of the broken road region descends with distance along the osculating circle to reduce the correlation between positions that are far apart. The decay function is described in the following equation:

$$DF(s, \kappa, \sigma) = exp(-\frac{s^2 + c\kappa^2}{\sigma^2}) \tag{3}$$

where $s$ denotes the arc length along the osculating circle, $\kappa$ represents the curvature, which can be computed after $v$, $\sigma$ is the scale parameter, and $c$ controls the decay of the field with curvature, which can be optimally adjusted as a function of the scale parameter $\sigma$. The expression is described as Equation (4).

$$c = \frac{-16 \ln(0.1)x(\sigma - 1)}{\pi^2} \tag{4}$$

The vote SV cast by a stick tensor $T$ at position $v$ is then expressed as Equation (5):

$$SV(T, \mathbf{v}) = \begin{cases} DF(v)R_{2\theta}TR_{2\theta}^T & if \ -\pi/4 \leq \theta \leq \pi/4 \\ 0 & otherwise \end{cases} \tag{5}$$

where $R_{2\theta}$ is a rotation matrix for an angle $2\theta$. The voting tensor is rotated and scaled following the decay function to produce the vote at a position in space.

- Ball votes

The ball voting field can be computed by integrating the votes of a rotating stick. It is assumed that $S(\phi, \omega)$ is a unitary stick tensor oriented in the direction $(1, \phi, \omega)$ in polar coordinates. Let this tensor have two degrees of freedom in its orientation that represent $\phi$ and $\omega$. The vote cast by a ball $B$ can then be described as follows:

$$BV(B, \upsilon) = \frac{3\lambda}{4\pi} \int_\Gamma SV(S(\phi, \omega), v)d\Gamma \tag{6}$$

where $\Gamma$ is the surface of a unitary sphere, $\lambda$ is any of the eigenvalues of $B$, and SV is defined in Equation (5).

- Tensor decomposition

As shown in equitation (2), refined tensors are decomposed into stick and ball components by a general saliency tensor. To improve the connectivity of the RDRCNN outputs, the curvature is expressed by $e_1 e_1^T$ for the tangent orientation by $\lambda_1 - \lambda_2$ for curve saliency.

- Voting collection and constraints with the RDRCNN results

Votes are collected by tensorial addition, which is equivalent to adding the matrix representations of the votes. The outputs and salient features are added and subtracted to detect the location of broken roads. Then, a morphology algorithm is used to thicken the outputs to the same size as their neighborhood.

## 3. Results

To verify the effectiveness of the proposed method, extensive experiments to extract roads from remote sensing images have been conducted on two datasets. We compare the proposed method with the same CNN architecture with a postprocessing stage and other CNN architectures. In this section, the experimental setup and results are illustrated.

### 3.1. Dataset Descriptions

### 3.1.1. Massachusetts road dataset

The Massachusetts road dataset consists of 60 training, 6 validation and 10 testing images (Table 3). The size of each image is 1500 × 1500 pixels with a spatial resolution of 1 m per pixel, composed of red, green and blue channels. This dataset was collected from Mnih [22] aerial images. The ground truth of the images consists of two classes: roads and non-roads.

**Table 3.** GF-2 satellite parameters.

| Item | Contents |
| --- | --- |
| Camera model | Panchromatic orthographic; Panchromatic front-view and rear-view; multispectral orthographic |
| Resolution | Subsatellite points full-color: 0.8 m; front- and rear-view 22° full color: 0.8 m; subsatellite points multispectral: 4 m |
| Wavelength | Panchromatic: 450 nm–900 nm Multispectral: Band1 (450 nm–520 nm); Band2 (520 nm–590 nm) Band3 (630 nm–690 nm); Band4 (770 nm–890 nm) |
| Revisit cycle | 5 days |

### 3.1.2. GF-2 Road Dataset

The raw images were obtained from the GF-2 satellite. The experimental area is located in Shenyang City. The details of the GF-2 satellite are described in Table 3.

The GF-2 road dataset, which we collected, was composed of roads and non-roads, including urban and rural regions. The raw image is 24750 × 20042 pixels; it has a spatial resolution of 0.8 m; and is composed of red, green, blue and infrared channels. This dataset roughly covers 317.465 km$^2$. The dataset was cut into 174 patches, and the size of all images in this dataset is 1500 × 1500 pixels. The dataset is divided into three subsets, i.e., training (60), validation (16) and testing (10), with no overlapping regions (Table 4).

**Table 4.** An overview of the datasets.

| Datasets | Training | Validation | Testing |
| --- | --- | --- | --- |
| Massachusetts Road Dataset | 60 | 6 | 10 |
| GF-2 Road Dataset | 60 | 16 | 10 |

### 3.1.3. Data Processing and Augmentation

In the GF-2 road dataset, due to the resolution difference between panchromatic and multispectral images, the pan-sharp algorithm [38] is used to obtain high-resolution color maps.

For training, images of both datasets are randomly cropped and augmented by random rotation. Then, we balance the samples by the number of road pixels in the two datasets.

### 3.2. Experimental Settings

### 3.2.1. Hyperparameter Settings

All experimental parameters for our method are chosen after conducting extensive experiments with various values and selecting the ones with good performance. Training is carried out by

optimizing the binary cross entropy loss function using the Adam algorithm [39]. Therefore, in the top of our network, sigmoid activation is used to sort the results. $\hat{y}$ is the distribution of the predicted result, $y$ is the distribution of the corresponding label image, and m is the total number of training images. Then, the loss function can be defined as:

$$loss(\hat{y}, y) = -1/m \sum_{i=1}^{m} y^{(i)} log \hat{y}^{(i)} + (1 - y^{(i)} log(1 - \hat{y}^{(i)})) \tag{7}$$

With moments of 0.9 and 0.999 corresponding to belta1 and belta2, the network was trained with an initial learning rate of 0.0001, which was reduced by a factor of 0.1 every 20 epochs. Figure 6 shows the loss function in each of the training and validation datasets after 100 epochs (epoch: each time the entire dataset is run). It is obvious that the error gradually decreases.

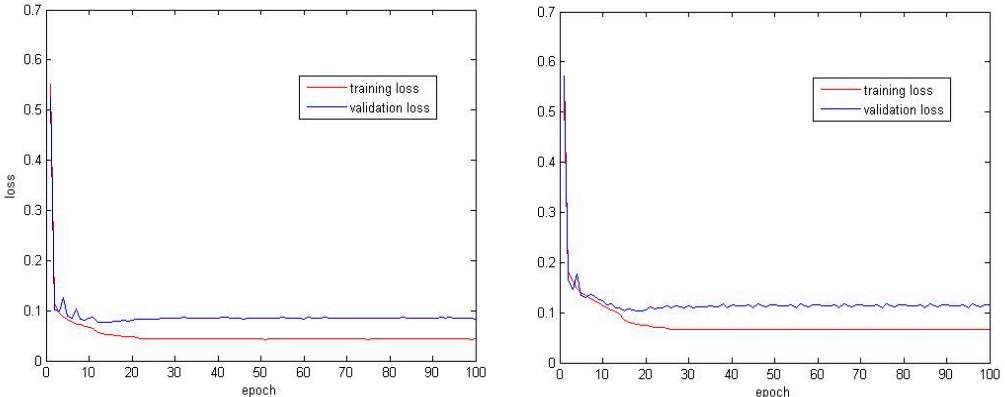

**Figure 6.** Training loss and validation loss of RDRCNN from two datasets. **Left**: Massachusetts road dataset. **Right**: GF-2 road dataset.

### 3.2.2. Training Environment Description

The module of the deep residual convolutional neural network is an end-to-end architecture, and the training patch is $256 \times 256 \times 3$. All experiments in this paper were performed on the deep learning framework Keras using the TensorFlow backend. The proposed method is implemented using a PC with an NVIDIA GTX 1070i and 8 GB of onboard memory.

### 3.2.3. Evaluation Metrics

To assess the quantitative performance of the proposed method in road network extraction, precision (P) (complexness), recall (R) [40] (correctness), F1 score, intersection over union (IoU) [41] and overall accuracy (OA) are used as metrics. The F1 [42] score is calculated by P and R. This score is a powerful evaluation metric for the harmonic mean of P and R and can be calculated as follows:

$$F_1 = 2 \times \frac{P \times R}{P + R} \tag{8}$$

where

$$P = \frac{TP}{TP + FP}, R = \frac{TP}{TP + FN} \tag{9}$$

where $R$ measures the proportion of matched pixels in the ground truth and $P$ is the percentage of matched pixels in the extraction results. *TP, FP, TN* and *FN* represent the number of true positives, false positives, true negatives and false negatives, respectively. *OA* measures the precision of roads and non-roads at the pixel level and can be represented as follows.

$$OA = \frac{TP + FP}{TP + FP + TF + TN} \tag{10}$$

In general, IoU is the ratio of the overlapping area of ground truth and predicted area to the total area. However, in the task of road extraction, IoU can be represented as follows.

$$IoU = \frac{TP}{TP + FP + FN} \tag{11}$$

The metrics mentioned above can be calculated using pixel-based confusion matrices.

### 3.3. Results and Analysis

To verify the robustness of the RDRCNN algorithm, we select two challenging datasets, the Massachusetts dataset [22] and the GF-2 dataset, for experiments. These datasets are optical remote sensing imagery datasets obtained from aerial and satellite imagery. To demonstrate the feasibility of the proposed method, we compare our method with CNN [22], U Net [26] and GL-Dense-U-Net [40]. For visual interpretation and analysis of the extraction results generated by different algorithms, the correctly classified road pixels are white, and the correctly classified nonroad pixels are black. If there are erroneous extractions and misclassified pixels, the corresponding pixels will be highlighted in red and blue, as shown in Figure 7.

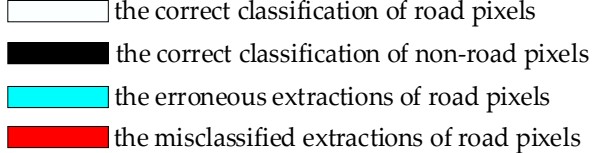

**Figure 7.** The legends for Figures 8–11.

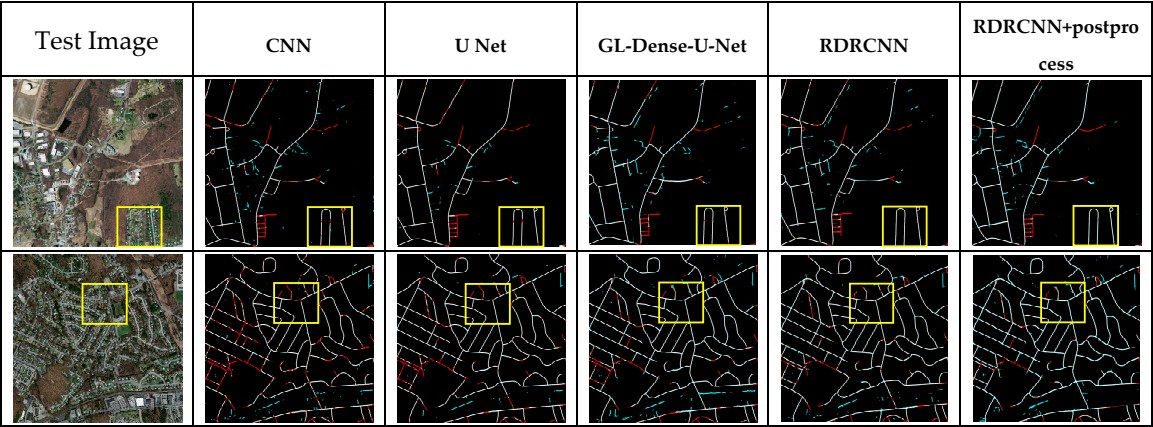

**Figure 8.** Comparison of the road extraction results of four methods in different experimental areas of the Massachusetts dataset. Zoomed-in view to see more details.

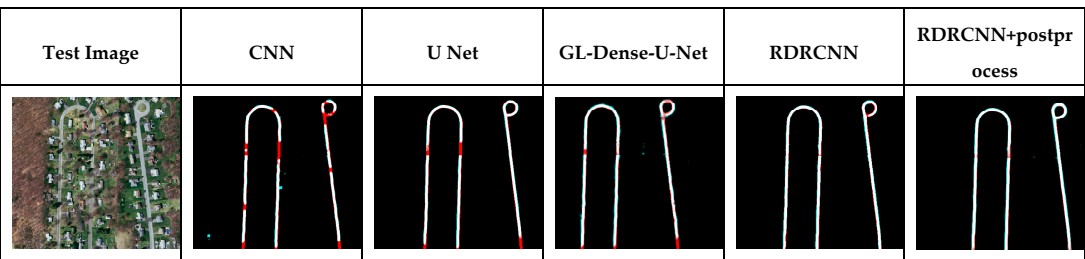

**Figure 9.** *Cont.*

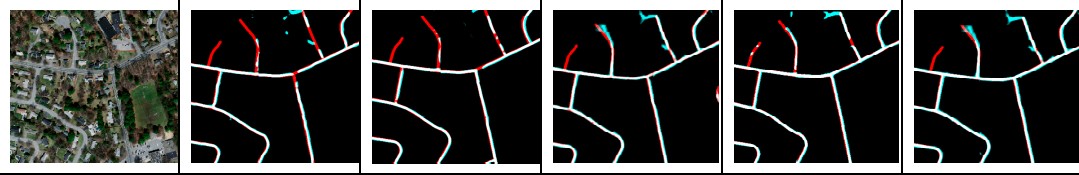

**Figure 9.** Comparison of the road extraction results of different algorithms in local areas (the small area in the yellow rectangular frame from the image in Figure 8).

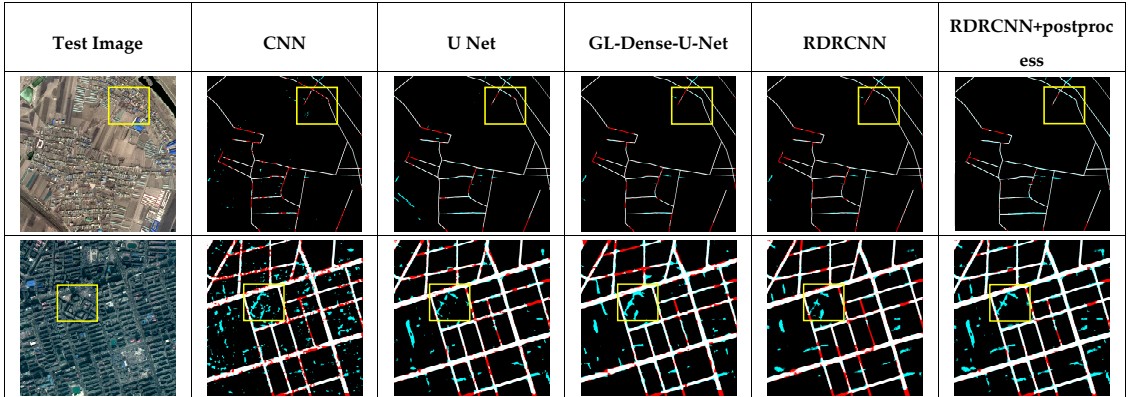

**Figure 10.** Comparison of the road extraction results of four methods in different experimental areas of the GF-2 dataset.

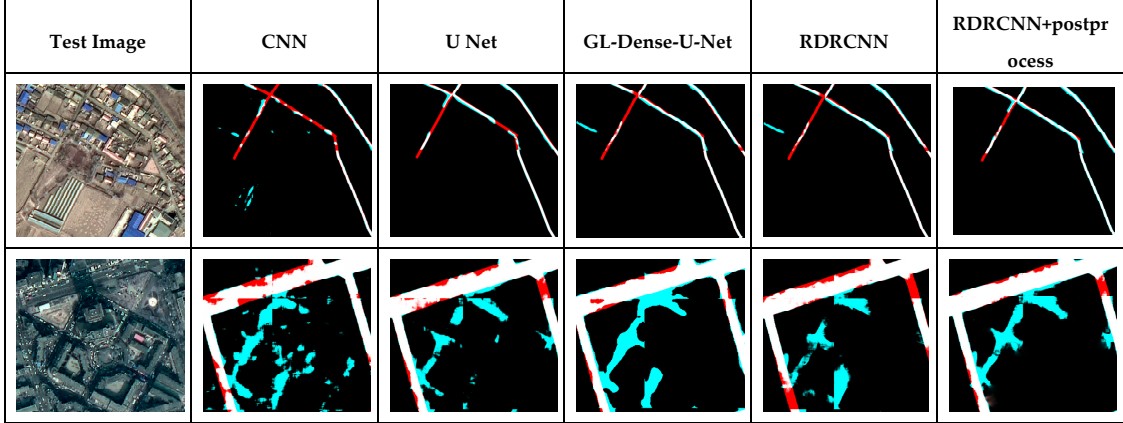

**Figure 11.** Comparison of road extraction results among different algorithms in local areas (the small area in the yellow rectangular frame from the image in Figure 10).

3.3.1. Massachusetts Data

Figure 8 presents two examples from the Massachusetts data after comparison with different methods. The proposed method achieves better performance in detail after postprocessing, as shown in Figure 8, and the metrics are described in Table 4. To facilitate observation and analysis, we selected a small area in the yellow rectangular frame from the image, and the extraction results are shown in Figure 9.

According to the results with the metrics shown in Table 5, the OA of the proposed method is clearly higher than that of the model using other CNN architectures, while the road extraction F1 scores and IoU have improved, with averages of 99.66%, and 99.32%, respectively. The additional postprocessing can clearly improve the performance for the extraction of broken roads and reduce the misclassification of pixels. Because the original remote sensing image should be clipped to train the deep neural convolutional network, in this paper, the RCU and the DPU are used to enlarge the FOV

and obtain the local and global road features. However, multiple scales are combined in different units in the deep feature extraction part. The improvements helped recover the road (by using the extracted information) and classify roads of different sizes.

**Table 5.** Comparison of the results in Figure 8 with different methods.

| Methods | Image Up | | | | | Image Down | | | | |
|---|---|---|---|---|---|---|---|---|---|---|
| | OA(%) | P(%) | R(%) | F1(%) | IoU(%) | OA(%) | P(%) | R(%) | F1(%) | IoU (%) |
| CNN | 97.56 | 97.62 | 99.96 | 98.72 | 97.47 | 94.91 | 94.93 | 99.90 | 97.35 | 94.84 |
| U Net | 97.53 | 97.53 | 99.98 | 98.74 | 97.51 | 94.78 | 94.75 | 99.95 | 97.29 | 94.72 |
| GL-Dense-U-Net | 97.85 | 99.99 | 98.86 | 99.43 | 98.86 | 94.85 | 99.99 | 98.05 | 99.01 | 98.05 |
| Ours | 98.13 | 99.99 | 99.15 | 99.57 | 99.15 | 95.07 | 99.99 | 99.03 | 99.51 | 99.03 |
| Ours+postprocess | 98.50 | 99.99 | 99.36 | 99.67 | 99.34 | 97.11 | 99.99 | 99.32 | 99.65 | 99.30 |

Table 6 shows the performance of the images in the Massachusetts dataset according to OA, completeness, correctness, F1 score and IoU at breakeven points. The main differences between the proposed method and other approaches include the sizes of the kernel filters, the number of filter units, the number of convolution layers, the depth of the network architecture and the postprocessing stage. Our method can achieve good performance because DPU is an effective way to enlarge the FOV without the loss of feature resolution. RCU can deepen the network configuration with the residual computed by the up and down layers, which proves the theory reported in [43] that the network configuration becomes deeper as it yields more accurate results. A statistical analysis of Table 6 indicates that the classification accuracy of RDRCNN with postprocessing is greater than those of CNN, U Net and GL-Dense-U-Net. The proposed method exhibits the greatest classification accuracy, with an IoU of 67.10%; the CNN exhibits the lowest classification accuracy, with an IoU of 59.69%. As shown in the schedule, we use the detailed statistics of the confusion matrix to describe the classification accuracy of the four methods in different experimental areas.

**Table 6.** Comparisons of the proposed and other deep-learning-based road extraction methods on the Massachusetts road test dataset in terms of better performance of testing.

| Dataset | Methods | OA (%) | P (%) | R (%) | F1 (%) | IoU (%) |
|---|---|---|---|---|---|---|
| Massachusetts | CNN [22] | 96.97 | 74.69 | 71.87 | 72.36 | 56.69 |
| | U Net [26] | 97.26 | 76.91 | 74.00 | 74.66 | 59.57 |
| | GL-Dense-U-Net [40] | 97.10 | 81.82 | 70.47 | 75.72 | 60.93 |
| | RDRCNN | 97.37 | 84.64 | 75.33 | 79.72 | 66.28 |
| | RDRCNN +postprocess | 98.01 | 85.35 | 75.75 | 80.31 | 67.10 |

### 3.3.2. GF-2 Data

The proposed method is applied to the GF-2 data, as shown in Figure 10, and the yellow rectangles are zoomed in to show more details, as shown in Figure 11. We selected two images in rural areas and urban areas.

The test image in the first row in Figure 10 covers rural areas. The rural roads are narrow and similar in texture to arable land. The image in the second row of Figure 10, as shown by the zoomed in image in Figure 11, covers the city range.

Road extraction is challenging due to the complexity of scenes. Because of the high resolution, the noise included cars, pedestrians and building shadows, which affect the detection results and are the major source of difficulty in the field of road extraction. Table 7 shows the comparison of the road extraction accuracy of different approaches with the same experimental configurations. A statistical analysis of Table 7 indicates that the extraction accuracy of our proposed method with the postprocessing stage is greater than that of the CNN and U Net methods. Our approach exhibits the

greatest classification accuracy in the two test images, with an average OA of 96.94%; the CNN exhibits the lowest classification accuracy, with an average OA of approximately 90.86%; and the U Net and the GL-Dense-U-Net fall in between, with average OAs of approximately 96.00% and 95.20%, respectively. The visual interpretation of Figures 10 and 11 indicates that, although the proposed method achieves a better result, the extraction of urban roads is more difficult than that of rural roads.

**Table 7.** The statistics of the accuracy of the extracted results in Figure 10 with different methods.

| Methods | Image Up | | | | | Image Down | | | | |
|---|---|---|---|---|---|---|---|---|---|---|
| | OA (%) | P (%) | R (%) | F1 (%) | IoU (%) | OA (%) | P (%) | R (%) | F1 (%) | IoU (%) |
| CNN | 97.12 | 97.12 | 99.99 | 98.54 | 97.12 | 84.60 | 84.19 | 99.89 | 91.37 | 84.11 |
| U Net | 98.51 | 98.66 | 99.87 | 99.26 | 98.53 | 93.48 | 93.83 | 98.49 | 96.10 | 92.49 |
| GL-Dense-U-Net | 97.05 | 97.69 | 99.22 | 98.45 | 96.95 | 92.90 | 94.94 | 96.24 | 95.58 | 91.53 |
| Our Network | 98.58 | 98.77 | 99.78 | 99.27 | 98.55 | 93.75 | 94.47 | 98.08 | 96.24 | 92.75 |
| Ours+postprocess | 98.99 | 99.39 | 99.57 | 99.48 | 98.96 | 94.88 | 96.92 | 96.81 | 96.86 | 93.91 |

In terms of the accuracy analysis for all images in the test set, Table 8 shows the metrics calculated from the test set at the breakeven point. The proposed method with postprocessing ranks first (with a precision of 82.41%), followed by U Net (with a precision of 79.74%), GL-Dense-U-Net (with a precision of 76.24%), and CNN (with a precision of 74.60%).

**Table 8.** Comparisons of the proposed and other deep-learning-based road extraction methods on the GF-2 road dataset in terms of better performance of testing.

| Dataset | Methods | OA (%) | P (%) | R (%) | F1 (%) | IoU (%) |
|---|---|---|---|---|---|---|
| | CNN | 96.34 | 74.60 | 70.19 | 70.89 | 54.91 |
| | U Net | 97.35 | 79.74 | 77.55 | 77.66 | 63.48 |
| GF-2 | GL-Dense-U-Net | 96.65 | 76.24 | 74.50 | 75.36 | 60.46 |
| | RDRCNN | 97.42 | 80.87 | 80.87 | 78.16 | 64.15 |
| | RDRCNN +post | 98.20 | 82.41 | 80.13 | 78.58 | 64.72 |

Math morphology and TV algorithms are used as postprocessing methods to further improve the performance. By considering the dependencies between the nearby pixels across patches, the TV algorithm is improved. Multiscale math morphological open operators are used to reduce the number of local misclassified pixels. Therefore, considering the connectivity of roads significantly improves the performance of the proposed method, especially at broken road regions and the boundaries of road intersections.

## 4. Discussion

The depth of the network architecture and the width of the FOV play significant roles in extracting roads from complex remote sensing imagery. In terms of high-resolution imagery, it is vital to enlarge the FOV to include high-resolution features. Thus, we propose two basic units, i.e., RCUs and DPUs, which help improve the ability to extract deep features for good performance. Profiting from the two effective units, the overall, precision, recall and F1 have been improved. To facilitate the effect of these units, we compare the results of the proposed model with either RCU or DPU and both cases. With the same training datasets and environment, the metrics are depicted in Table 9.

**Table 9.** Comparison of the results with either RCUs or DPUs at the breakeven point.

| Datasets | Units | OA (%) | P (%) | R (%) | F1 (%) | IoU (%) |
|---|---|---|---|---|---|---|
| Massachusetts Road Dataset | Only RCU | 97.18 | 77.00 | 72.91 | 74.03 | 58.77 |
| | Only DPU | 97.20 | 76.14 | 73.92 | 74.18 | 58.96 |
| | Both | 97.37 | 84.64 | 75.33 | 79.72 | 66.28 |
| GF-2 Road Dataset | Only RCU | 97.00 | 78.36 | 72.89 | 74.05 | 58.79 |
| | Only DPU | 97.00 | 77.88 | 73.65 | 74.45 | 59.30 |
| | Both | 97.42 | 80.87 | 80.87 | 78.16 | 64.15 |

OA improved by an average of 0.67% and 0.66% when the structure included only RCU and DPU, respectively. Meanwhile, the F1 scores for road detection improved by an average of 2.78% and 2.5%, respectively. RCU is used to create a deeper network to gain more road information, while DPU enlarges the FOV without losing feature resolution information. The usage of both units improved road extraction from high-resolution imagery.

Because none of the units (RCU or DPU) have resolution loss, the number of pooling layers is chosen according to the spatial resolution of the input image to be reasonably high to facilitate over-segmentation. Road width is relatively small in high-resolution images ($3 \leq \omega \leq 8$, where $\omega$ is the minimize size of a road). Assuming that the spatial resolution of an image is $r$, the number of pooling layers is $n$, and the deep characteristic resolution is $r_d$, the relationship can be depicted as follows.

$$r_d = 2^{n-1} \cdot r \leq \omega \tag{12}$$

Thus, if the spatial resolution of the input image is 0.8 m, where 3–8 m corresponds to 4–10 pixels, the number of pooling layers can be computed with Equation (12) to be between $n = 2$ and $n = 4$. To facilitate over segmentation, $n = 3$ is selected as the optimal parameter value. We also did some experiments. When $n = 2$, the training accuracy is suboptimal. When $n = 4$, the model is overfitted.

From the abovementioned figures in this paper, some issues remain to be considered, such as the misclassification pixels and the unclear edge regions in urban areas. With the development of remote sensing technology, it has become much easier to gain high-resolution remote sensing imagery, which poses a major challenge for image interpretation. In this paper, we proposed a novel method for solving the problem of broken roads, which improved the image road classification performance. However, the datasets do not contain images from different sensors. Therefore, the trained model cannot represent the optimal result. In the future, a new deep neural network is required to extract roads from imagery with different sources. Moreover, based on the performance in this paper, the model can be added with other constraints, such as the shape features, to improve pattern recognition. Therefore, our future work will focus on extracting more accurate maps from complex scenes and updating the database of roads to detect changes over time.

## 5. Conclusions

In this paper, a novel method for extracting roads in high-resolution remote sensing imagery acquired over urban and rural areas is proposed. The method consists of two major stages: RDRCNN and postprocessing with a TV algorithm. The RDRCNN model is based on ResNet and U Net architectures with dilated convolution operators. Line salient features of roads are integrated with RDRCNN outputs to connect broken regions. The major contributions of this paper are the refinement of the CNN architecture to detect road regions more accurately and the use of the blind voting method to join misclassification regions because of visual occlusion. The method is evaluated using two challenging datasets, one of which was collected by our team. This method is also compared with other methods. The experiments prove that the method achieves good performance in the task of road extraction from complex backgrounds (city and countryside), but it requires additional processing to more accurately outline boundaries, especially in urban areas.

**Author Contributions:** Conceptualization, L.G.; Funding acquisition, W.S.; Methodology, L.G. and J.D.; Project administration, W.S.; Resources, W.S.; Writing—original draft, L.G.; Writing—review & editing, L.G. and Y.C.

**Funding:** This work was supported by the public welfare research fund in Liaoning Province (NO. 20170003).

**Acknowledgments:** The authors wish to thank the editors and reviewers.

**Conflicts of Interest:** The authors declare no conflicts of interest.

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
