# Peer review of "Road Extraction from High-Resolution Remote Sensing Imagery Using Refined Deep Residual Convolutional Neural Network"

_remotesensing, doi:10.3390/rs11050552_

Round 1

Reviewer 1 Report

In this paper, Authors propose a novel method for extracting roads from optical satellite images using a refined deep residual convolutional neural network (RDRCNN) with a postprocessing stage. The method is very interesting and well described. The scientific problem is correctly presented and discussed. The level of the article is high and deserves to be published without any corrections. I did not notice any mistakes or omissions.  One small question. My remark concerns the use of GF-2 satellite images. Spatial resolution of panchromatic images is 0.8 meters and multispectral 3.24. The article examines color images with a resolution of 0.8. In that case, they were probably subjected to a pansharpening process. Question - how this process (its incorrect) influenced the accuracy of the method?

Author Response

Dear Reviewer:

      Thank you for your comments concerning our manuscript entitled “Road Extraction from High-resolution Remote Sensing Imagery using Refined Deep Residual Convolutional Neural Network” (ID: remotesensing-430147). These comments are all valuable and very helpful for revising and improving our paper, as well as the important guiding significance to our researchers. We have studied comments carefully and have made correction. Revised portion are marked in red in the paper. The main corrections in the paper and the responds to the comments are as following.

Response to comment: how this process (its incorrect) influenced the accuracy of the method?

Response: Yes, the process may influence the accuracy of the method. But it is slight. If we only use the panchromatic images, the accuracy of test images is unsatisfied. As analyzing, it may due to the lack of spectral information. The convolutional neural network needs both spectral and spatial features. Thus, we augment Section 3.1.3 Data processing and augmentation in Line 229.

We tried our best to improve the manuscript and made some changes in the manuscript. These changes will not influence the content the framework of the paper and here we did not list the changes but marked in red in revised paper. We appreciate for you warm work earnestly, and hope that the correction will meet with approval.

Once again, thank you very much for your comments and suggestion.

Yours Lynn Gao

Reviewer 2 Report

In this paper, the authors propose a road extraction method from high-resolution remote sensing imagery by RDRCNN. The work is well organized and the experimental results are better than the compared methods. However, I have some questions in the following.

1. The depth  of the network architecture and the width of the FOV play significant roles in this work. How many RCU and DPU are used in the proposed method and how to determine the quantity?

2. Could the author do some analyses with the quantity of RCU and DPU?

3. In line 238, there is a text mistake, figure 7 must be figure 6.

Author Response

Dear Reviewer:

      Thank you for your comments concerning our manuscript entitled “Road Extraction from High-resolution Remote Sensing Imagery using Refined Deep Residual Convolutional Neural Network” (ID: remotesensing-430147). These comments are all valuable and very helpful for revising and improving our paper, as well as the important guiding significance to our researchers. We have studied comments carefully and have made correction. Revised portion are marked in red in the paper. The main corrections in the paper and the responds to the comments are as following.

Response to comment: The depth of the network architecture and the width of the FOV play significant roles in this work. How many RCU and DPU are used in the proposed method and how to determine the quantity?

Response: In line 371, which in the section of Discussion, we add the analysis of the numbers of RCU and DPU used in the proposed method. And we build a simple model between them to ensure the optimal architecture.

Response to comment: Could the author do some analyses with the quantity of RCU and DPU?

Response: The quantity of RCU and DPU determines the depth of the network. As the equation (12) in line 377, we do the analyses of the depth of network with RCU and DPU as the basic units. Every unit contains of an RCU or DPU, otherwise, the deeper of the architecture, the more possible it may be overfitting.

Response to comment: In line 238, there is a text mistake, figure 7 must be figure 6.

Response: we are very sorry for our incorrect writing in line 238. We have corrected the Figure 6.

We tried our best to improve the manuscript and made some changes in the manuscript. We appreciate for you warm work earnestly, and hope that the correction will meet with approval.

Once again, thank you very much for your comments and suggestion.

Yours Lynn Gao

Reviewer 3 Report

The authors suggest a method for road extraction in satellite imagery based on modified deep residual network and post processing. I have several concerns about the results:

There is a huge gap between training and validation curves (Figure 6) for both dataset that shows strong overfitting. I cannot therefore trust their results. Also, validation curve is quite oscillatory.

Moreover, I would like to see the results in comparison with the recent techniques that were developed for similar imagery for road extraction. Both references [19] 1nd [22] are quite old. 

Can you compare your results with reference [32]: Y. Xu, Z. Xie, Y. Feng, and Z. Chen, “Road Extraction from High-Resolution Remote Sensing Imagery Using Deep Learning,” Remote Sensing, vol. 10, no. 9, p. 1461, Sep. 2018.

And/or 

“recurrent neural networks to correct satellite image classification maps” by Maggiori et. Al.

I would like to see the improvement of their methods in comparison to similar methods in literature.

IOU (intersection over Union) is also a better metric for semantic segmentation. Can you measure IOU in comparison to other techniques?

What are your contribution in comparison to the recent techniques in the literature on road extraction. This is not clear to me.

Author Response

Dear Reviewer:

      Thank you for your comments concerning our manuscript entitled “Road Extraction from High-resolution Remote Sensing Imagery using Refined Deep Residual Convolutional Neural Network” (ID: remotesensing-430147). These comments are all valuable and very helpful for revising and improving our paper, as well as the important guiding significance to our researchers. We have studied comments carefully and have made correction. Revised portion are marked in red in the paper. The main corrections in the paper and the responds to the comments are as following.

Response to comment: There is a huge gap between training and validation curves (Figure 6) for both dataset that shows strong overfitting. I cannot therefore trust their results. Also, validation curve is quite oscillatory.

Response: Due to the comment that the curves show strong overfitting, we augment three parts in our manuscript, where in line 129, line 232, and line 244, respectively. To prevent overfitting, in line 129, we add the batch normalized layer to the bottom of the basic unit. In line 232, images of both datasets are randomly cropped and augmented by random rotation. Then, we balance the samples by the number of road pixels in the two datasets. Meanwhile, in line 244, we revise the optimal hyper parameter, which an initial learning rate of 0.0001 reduced by a factor of 0.1 every 20 epochs. As shown in Figure 6 of our manuscript, we compare the training loss and validation loss of the two datasets.

Response to comment:  compare your results with reference [32]: Y. Xu, Z. Xie, Y. Feng, and Z. Chen, “Road Extraction from High-Resolution Remote Sensing Imagery Using Deep Learning,” Remote Sensing, vol. 10, no. 9, p. 1461, Sep. 2018.

Response: In the experiments section, we augment the comparison method of the both datasets with GL-Dense-U-Net.

Response to comment: Can you measure IOU in comparison to other techniques?

Response: In the section 3.2.3 and section 3.3, we augment the IoU measure in comparison to the other methods.

Response to comment:  What is your contribution in comparison to the recent techniques in the literature on road extraction. This is not clear to me.

Response: In line 81, we rewrite the three contributions of our manuscript.

We tried our best to improve the manuscript and made some changes in the manuscript. We appreciate for your warm work earnestly, and hope that the correction will meet with approval.

Once again, thank you very much for your comments and suggestion.

Yours Lynn Gao

Round 2

Reviewer 2 Report

The authors have revised the paper according to the reviewer's comments. However, the text error is still not corrected in line 244. "Figure 7" must be corrected as "Figure 6".

Author Response

Thank  for the comments concerning our manuscript entitled “Road Extraction from High-resolution Remote Sensing Imagery using Refined Deep Residual Convolutional Neural Network” (ID: remotesensing-430147).  The revised portion is marked in red in the paper.

Reviewer 3 Report

Thank you for revising the manuscript. I am happy with the changes.

Author Response

Thank you!

This manuscript is a resubmission of an earlier submission. The following is a list of the peer review reports and author responses from that submission.